# The Role of the *PAA1* Gene on Melatonin Biosynthesis in *Saccharomyces cerevisiae*: A Search of New Arylalkylamine *N*-Acetyltransferases

**DOI:** 10.3390/microorganisms11051115

**Published:** 2023-04-25

**Authors:** Ricardo Bisquert, Andrés Planells-Cárcel, Javier Alonso-del-Real, Sara Muñiz-Calvo, José Manuel Guillamón

**Affiliations:** 1Instituto de Agroquímica y Tecnología de Alimentos IATA, CSIC, 46980 Paterna, Spain; r.bisquert@iata.csic.es (R.B.); anplacar@iata.csic.es (A.P.-C.); javier.alonso@iata.csic.es (J.A.-d.-R.); sara.muniz@iata.csic.es (S.M.-C.); 2Instituto de Biomedicina de Valencia IBV, CSIC, 46010 Valencia, Spain; 3Department of Life Sciences, Chalmers University of Technology, SE41296 Gothenburg, Sweden

**Keywords:** yeast, *E. coli*, melatonin, tryptophan metabolism, *N*-acetylserotonin, 5-methoxytryptamine, polyamine acetyltransferase, *HPA2*, AANAT activity

## Abstract

Recently, the presence of melatonin in fermented beverages has been correlated with yeast metabolism during alcoholic fermentation. Melatonin, originally considered a unique product of the pineal gland of vertebrates, has been also identified in a wide range of invertebrates, plants, bacteria, and fungi in the last two decades. These findings bring the challenge of studying the function of melatonin in yeasts and the mechanisms underlying its synthesis. However, the necessary information to improve the selection and production of this interesting molecule in fermented beverages is to disclose the genes involved in the metabolic pathway. So far, only one gene has been proposed as involved in melatonin production in *Saccharomyces cerevisiae*, *PAA1*, a polyamine acetyltransferase, a homolog of the vertebrate’s aralkylamine *N*-acetyltransferase (AANAT). In this study, we assessed the in vivo function of *PAA1* by evaluating the bioconversion of the different possible substrates, such as 5-methoxytryptamine, tryptamine, and serotonin, using different protein expression platforms. Moreover, we expanded the search for new *N*-acetyltransferase candidates by combining a global transcriptome analysis and the use of powerful bioinformatic tools to predict similar domains to AANAT in *S. cerevisiae*. The AANAT activity of the candidate genes was validated by their overexpression in *E. coli* because, curiously, this system evidenced higher differences than the overexpression in their own host *S. cerevisiae.* Our results confirm that *PAA1* possesses the ability to acetylate different aralkylamines, but AANAT activity does not seem to be the main acetylation activity. Moreover, we also prove that Paa1p is not the only enzyme with this AANAT activity. Our search of new genes detected *HPA2* as a new arylalkylamine *N*-acetyltransferase in *S. cerevisiae*. This is the first report that clearly proves the involvement of this enzyme in AANAT activity.

## 1. Introduction

After the discovery of melatonin outside the animal kingdom, research on melatonin in other clades emerged. Thus, melatonin was found to be a ubiquitous phylogenetically ancient molecule in almost every organism, from primitive photosynthetic bacteria to humans [1]. For melatonin synthesis, the majority of studies have been performed in vertebrates, particularly in mammals, and more recently in plants [2]. The occurrence of melatonin in yeast was described for the first time by Sprenger et al. [3] as a product of the metabolism of precursors such as tryptophan, serotonin, *N-*acetylserotonin, and 5-methoxytryptamine. These results have been later confirmed and extended by numerous studies demonstrating yeast is responsible for the biosynthesis of this molecule in a fermentative context, bringing the challenge of studying the function of melatonin in yeast and the mechanisms underlying its synthesis [4,5,6,7]. Melatonin exerts multiple physiological roles on different organisms, from regulating biorhythms and aging, modulating immune system response, inhibiting tumor growth, and protecting from UV light, among others [8,9,10]. In the case of *Saccharomyces cerevisiae*, it has been empirically demonstrated that melatonin has a protective role against oxidizing agents and UV light [11,12,13]. Regarding the melatonin biosynthetic pathway, there is a high degree of conservation of the enzymatic reactions that lead to melatonin synthesis from tryptophan. It is the order of these reactions that characterizes the biosynthetic route in yeasts. *S. cerevisiae* seems to convert tryptophan to tryptamine in the first decarboxylation step, followed by hydroxylation to form serotonin. Then, melatonin is formed from serotonin by *N-*acetylation followed by *O*-methylation of *N-*acetylserotonin or, alternatively, by an *O*-methylation of serotonin to form 5-methoxytryptamine followed by its *N-*acetylation, which is the preferred alternative for *S. cerevisiae*, although further evidence suggests more branches on the pathway in which the tryptophan as a precursor is [14]. Therefore, the classical melatonin pathway model for vertebrates does not seem to apply to yeast.

Despite all the advances in melatonin biosynthesis in yeast, there is still uncertainty around the specific genes involved in the route. Only one gene has been described and characterized as involved in melatonin production, *PAA1*, a polyamine acetyltransferase, homolog of the vertebrate’s aralkylamine *N*-acetyltransferase (*AANAT*), which can acetylate serotonin to *N-*acetylserotonin and 5-methoxytryptamine to melatonin [15], while the remaining genes and enzymes of the route are still unknown. Therefore, the search for genes homologous to those described in vertebrates and plants in *S. cerevisiae* represents a challenging goal and a key point to improve the synthesis of these molecules during fermentation processes in which *S. cerevisiae* participates.

In this study, we assessed the in vivo function of *PAA1* by evaluating the bioconversion of the different possible substrates, such as 5-methoxytryptamine, tryptamine, and serotonin using different protein expression platforms. To that aim, we overexpressed the *PAA1* gene, and the aralkylamine *N-*acetyltransferase of *Bos taurus* (Bt*AANAT*) as a positive control, in *S. cerevisiae* and *Escherichia coli*, and we measured the production of acetylated metabolites after a precursor pulse into the media. As the results evidenced the presence of alternative enzymes with AANAT activity in *S. cerevisiae*, we expanded the search for *N-*acetyltransferase candidates by combining a global transcriptional expression analysis (RNAseq), under melatonin synthesis conditions, and the use of powerful bioinformatic tools. This strategy has allowed us to propose new candidates to explain melatonin-related acetylation activity in yeast.

## 2. Materials and Methods

### 2.1. Strains and Culture Media

Strain *E. coli* NZYa (NzyTech, Lisboa, Portugal) was used as a cloning host for plasmid construction and amplification. Strain *E. coli* Rosetta(DE3) competent cells (Novagen, Darmstadt, Germany) were used to enhance the expression of eukaryotic proteins that contain codons rarely used in *E. coli*. *E. coli* cells were cultured in LB medium containing 10 g·L^−1^ of tryptone, 5 g·L^−1^ of yeast extract, and 5 g·L^−1^ of NaCl supplemented with 100 µg·L^−1^ of ampicillin and 34 µg·L^−1^ chloramphenicol to maintain plasmids at 37 °C. A 2xTY medium consisted of 16 g·L^−1^ of tryptone, 10 g·L^−1^ of yeast extract, and 5 g·L^−1^ of NaCl.

Yeast strain BY4743 without plasmids was maintained and grown in YPD medium (20 g·L^−1^ glucose, 20 g·L^−1^ peptone, 10 g·L^−1^ yeast extract) whereas strains carrying plasmids were maintained and grown in SC without uracil (20 g·L^−1^ glucose, 1.7 g·L^−1^ yeast nitrogen base (YNB) without amino acids and ammonium sulfate (BD Difco, Sparks, MD, USA), 5 g·L^−1^ ammonium sulfate and 1.9 g·L^−1^ of SC-ura drop-out powder (Formedium, Swaffham, UK)), both supplemented with 16 g·L^−1^ agar (Condalab, Madrid, Spain) for solid media at 28 °C.

### 2.2. Plasmid Construction

The plasmids and primers herein used are listed in Table 1 and Table 2, respectively. Genes from *Saccharomyces cerevisiae PAA1*, *ARD1*, *NAT4*, *GNA1*, *YIR042C*, *HPA2*, *NAT3*, and *HAT1* were PCR amplified from genomic DNA of yeast strain BY4743 and Bt*AANAT* was amplified from the plasmid pCfB2628 [16]. For amplification, Phusion DNA polymerase (Thermo Scientific, Waltham, MA, USA) and the primer pairs AANAT F BamHI/AANAT R XhoI, PAA1 F BamHI/PAA1 R XhoI, NAT4 F BamHI/NAT4 R XhoI, GNA1 F BamHI/GNA1 R XhoI, YIR042C F BamHI/YIR042C R XhoI, HPA2 F BamHI/HPA2 R XhoI, and NAT3 F BamHI/NAT3 R XhoI, introducing a BamHI site and an XhoI site, were, respectively, used. In the case of *HAT1* and *ARD1* genes, HAT1 F EcoRI/HAT1 R XhoI and ARD1 F EcoRI/ARD1 R XhoI, introducing an EcoRI site and an XhoI site, were used. The resulting PCR fragments were digested with BamHI or EcoRI and XhoI and cloned next to the GPD promoter of the opened plasmid p426GPD [17] or pGEX-5X-1. The resulting plasmids (Table 1) were transformed into *E. coli* and the transformants were screened by colony PCR and sequenced using the pair primers GPDPro-F/CYC1-R and pGEX seq F/ pGEX seq R for p426GPD and pGEX-5X-1, respectively.

### 2.3. Bioconversion Assays

To overexpress heterologous genes in *E. coli*, the plasmids pGEX-5X-1 constructed with the different candidate genes (Table 1) were transformed into Rosetta™ (DE3) competent cells (Novagen), empty vector was also transformed and used as negative control. Transformants were grown under continuous shaking at 37 °C overnight in 15 mL tubes with 5 mL of LB medium, supplemented with 100 µg·L^−1^ ampicillin and 34 µg·L^−1^ chloramphenicol. The next day, 15 µL of grown preculture were inoculated into 1.5 mL of 2xTY medium with 1% glucose, supplemented with ampicillin and chloramphenicol at the same concentrations mentioned above. Cultures were grown at 37 °C until OD_600_ reached 0.6, after that, 0.25 mM of IPTG and 1 mM of the desired precursor were added to the culture. The culture was further grown for 24 h, at 28 °C under 300 rpm orbital shaking, then samples were taken and stored at −20 °C until extraction and HPLC analysis.

To overexpress the different genes in *S. cerevisiae*, a vector-based constitutive overexpression system was employed. To overexpress the genes of interest, high-copy number vector p426GPD, which enables strong constitutive expression by using GPD (*TDH3*) promoter, was used to clone the candidate genes and transformed them into BY4743 yeast strain (EUROSCARF, Oberursel, Germany). Empty vector was also transformed and used as a negative control in the assays. Individual transformants were grown overnight in 1.5 mL tubes containing 0.8 mL of SC without uracil (SC-ura) medium at 28 °C under continuous shaking at 150 rpm, then 30 µL of the grown pre-inoculum was inoculated into 1.5 mL of fresh SC-ura medium in a 24 well microtiter plate with 2 mL well capacity. Plates were incubated at 28 °C under constant shaking in a microplate orbital shaker at 300 rpm and 1 mM of precursor was added to the culture when late exponential phase (0.6 to 0.8 OD_600_) was reached. Samples were taken after 50 h and stored at −20 °C until extraction and analysis.

### 2.4. Drop Test

After growth on SC at 28 °C up to the stationary phase, the cells were harvested by centrifugation, washed with sterile water, resuspended in sterile water to an OD_600_ value of 0.5, and followed by serial dilution. From each dilution, 3.5 µL were spotted onto SC-ura with or without pantothenate agar plates. Plates were incubated at 28 for 2 and 5 days.

### 2.5. Gene Expression Analysis

To study global gene expression under melatonin synthesis conditions, preinocula of BY4743 were grown overnight in SC medium and inoculated in 250 mL shake flasks containing 100 mL of SC medium to an initial OD_600_ of 0.15 and incubated at 28 °C with 150 rpm orbital shaking. When cells reached OD_600_ of 0.8, a supplementation of 5-methoxytryptamine was added to a final concentration of 1 mM. Non-supplemented cultures were used as negative controls and 12.5 mL of each culture were taken as samples 15 and 45 min after supplementation. For each sample, cells were pelleted and snap-frozen with liquid nitrogen and stored at −80 °C for further RNA extraction. Supernatant was also collected to determine melatonin production under these experimental conditions by HPLC-MS/MS.

To assess the overexpression of *PAA1* in yeast we performed quantitative PCR (qPCR) on cells bearing plasmid p426GPD PAA1 and the wild-type control. Cells were grown overnight in SC-ura and SC media, respectively, then inoculated in shake flasks as described above. When they reached an OD_600_ of 0.8 cells were pelleted, snap-frozen, and stored at −80 °C until RNA extraction.

To extract RNA, cell pellets were resuspended in 0.4 mL of LETS buffer [0.1 M LiCl, 0.01 M EDTA, pH 8.0, 0.01 MTris-HCl, pH 7.4, and 0.2% (*w*/*v*) SDS] and added to 2 mL screw tubes containing 0.4 mL of phenol (pH 4.5)-chloroform (5:1) and 0.3 mL of glass beads. Then, cells were ruptured with a Tehtnica MillMix 20 homogenizer (Tehtnica, Zelezniki, Slovenia). Supernatants were extracted with phenol–chloroform (5:1) and chloroform-isoamyl alcohol (24:1). RNA was precipitated twice overnight at −20 °C, first by adding 2.5 volumes of 96% ethanol and 0.1 volume of 5 M LiCl, and secondly by adding 2.5 volumes of 96% ethanol and 0.1 volume of 3 M sodium acetate. RNA was finally resuspended in RNase-free MilliQ water, and the concentration was determined in a NanoDrop spectrophotometer (Thermo Scientific, USA).

RNA sequencing was performed by SCSIE (University of Valencia, Valencia, Spain). Briefly, RNA quality was determined with a Bioanalyzer 2100 (Agilent, Santa Clara, CA, USA), and concentration was measured with a Qubit^®^2.0 (Life Technologies, Carlsbad, CA, USA), yielding RNA integrity numbers (RIN) between 9.4 and 9.7, thus indicating nearly intact RNA, and concentrations ranged between 400 and 900 ng/mL across all samples. Libraries for RNA-Seq were prepared with TruSeq^®^ stranded mRNA Kit (Illumina, San Diego, CA, USA) following the manufacturer’s protocol and subsequently sequenced on an Illumina HiSeq 2500 with 2 × 75 bp paired-end reads. Adapters from raw reads were trimmed with *trimmomatic*. Adapter content and read quality were assessed with *FastQC*, then reads were mapped to the S288c genome with *Bowtie2,* and count tables were obtained with *htseq-count*. The gene expression abundance was normalized by log2-CPM (counts per million, corrected for the different library sizes, expressed in a log2 scale) using *edgeR* (v3.36.0). *limma* package (v3.50.3) was used to estimate log2-CPM mean variance, and establish the different contrasts of hypothesis, and an empirical Bayes moderation of the standard errors was applied to increase statistical power of differentially expressed genes (DEGs). Significance (*p* < 0.05) was accounted for on the corrected *p*-value obtained from the tests applied by *eBayes* module (*limma* package). Whole statistical computing was run on *R* software (v4.1.3).

Enrichment of GO terms and gene clustering was analyzed based on the identified DEGs. Specific gene functions and biological pathways were annotated according to SGD http://www.yeastgenome.org (accessed on 7 February 2023) and UniProt http://www.uniprot.org/ (accessed on 7 February 2023). The interaction networks of DEGs were obtained using the STRING v11.5 database http://string-db.org/ (accessed on 7 February 2023).

RNA used in qPCR analysis was treated for 15 min at 25 °C with DNase I RNase-free (Roche, Basil, Switzerland) according to the manufacturer’s recommendations using 1 µg of total RNA from each sample. NZY First-Strand cDNA Synthesis kit (NZTtech, Lisboa, Portugal) was used to synthesize cDNA from the DNase I-treated RNA following the manufacturer’s protocol. Quantitative real-time PCR was performed in a Light Cycler 480 II (Roche) using the SYBR Premix Ex Taq kit (TaKaRa, Shiga, Japan) for fluorescent labeling. For this purpose, 2.5 µL cDNA was added to each reaction at a final volume of 10 µL. The real-time PCRs were performed using 0.2 µM of the corresponding oligonucleotides under the following conditions: 95 °C for 10 s, followed by 40 cycles of 10 s at 95 °C and 15 s at 55 °C. At the end of the amplification cycles, a melting curve analysis was conducted to verify the specificity of the reaction. A standard curve was made with serial dilutions of the cDNA sample (2 × 10^−1^, 1 × 10^−1^, 2 × 10^−2^, 1 × 10^−2^, 2 × 10^−3^, 1 × 10^−3^). The primers used to determine the transcript levels are represented in Table 2.

### 2.6. Statistical Analysis

All the experiments were carried out at least in triplicate. Data were expressed as the mean values ± standard deviation. Experimental results were analyzed and compared by statistical analyses such as ANOVA, Tukey’s honestly significant difference (HSD), and t-tests using GraphPad Prism (GraphPad Software Inc., San Diego, CA, USA) software (v7.00). A confidence level of at least 95% was considered. Significance was graphically shown either as *p* < 0.05 (*), *p* < 0.005 (**), and *p* < 0.001 (***) or using letters “a”, “b”, and “c” to reflect significant differences. All statistical program packages involved in the RNAseq analysis were run on *R* software (v4.1.3) as described above.

## 3. Results and Discussion

Ganguly et al. [15] cloned and overexpressed the *S. cerevisiae PAA1* gene in *E. coli* and, after its purification, characterized its enzymatic activity in vitro. These authors concluded that this enzyme has activity generally typical for AANAT family members, although the substrate preference pattern was somewhat broader, the specific activity was lower, and the pH optimum was higher than the reference mammalian AANAT. Complementary to this previous study, we aim to characterize the in vivo function of *PAA1* by overexpressing this gene in *S. cerevisiae* and determine the impact of this overexpression on the acetylated products of the melatonin biosynthesis pathway.

### 3.1. Overexpression of PAA1 in S. cerevisiae

Conversely to our expectations, the overexpression of *PAA1* in *S. cerevisiae*, in a medium supplemented with 5-methoxytryptamine, as this precursor has been described as the best amine substrate for *PAA1* [15], did not show significant differences in melatonin production in comparison to the wild-type strain transformed with the empty vector (control strain). Nonetheless, Bt*AANAT* overexpression produced a 25-fold higher melatonin concentration than the control and the *PAA1* overexpressing strain (PAA1) (Figure 1A). In view of this unexpected result, we also tested the acetylase capacity of *PAA1* on other substrates (tryptamine and serotonin). In this assay, we also included the null mutant strain (paa1) to evaluate a possible loss of acetylating function for any of the substrates. Again, no significant differences in acetylated products from any substrate were detected among these strains. Null mutants still showed the same levels of acetylated products as the control or the overexpressed strains (Figure 1B). This result contradicts the previous Ganguly et al. [15] study, which reported a reduced arylalkylamine acetylation in crude homogenates of the paa1 mutant strain in comparison with the wild type.

In order to assure that the induction of *PAA1* in the overexpressing strain had been achieved, we determined the gene transcriptional activity of both the control and overexpressing *PAA1* strain by qPCR. *PAA1* was found to be expressed over 270 time-fold in the PAA1 strain compared to the control strain, which indicates that overexpression had been correctly achieved (Figure 1C). As the lack of increases in acetylation activity was not connected with the transcriptional activity, we tested if functional overexpressed proteins were also obtained. Liu et al. [18] reported that *PAA1* overexpression caused partial growth inhibition in a medium without pantothenate, but not in a rich medium. An increase in intracellular Paa1p led to an excess of acetylated polyamines, such as putrescine, spermidine, and spermine, which are the immediate precursors in the synthesis of pantothenate (Appendix A). A shortage in these polyamines turned out in a reduction of intracellular pantothenate and a growth defect in the absence of this vitamin in the growth medium. In turn, the FMS1 gene encodes a polyamine oxidase that converts spermine into 3-aminopropanal, which is then converted to β-alanine. The β-alanine is a precursor of pantothenate, which is in turn a precursor of coenzyme A (Appendix A). Based on these previous metabolic data, we performed a drop test in an SC medium with and without pantothenate to indirectly verify if we had functionally overexpressed *PAA1*. We also tested the *FMS1* null mutant strain (fms1) which is unable to grow in the absence of pantothenate. These drop tests showed a growth defect in the PAA1 strain and a total inhibition in the fms1 strain, as previously reported [18]. This result evidenced that *PAA1* was not only overexpressed but also translated, into functional proteins (Figure 1D).

In light of these results, the absence of significant arylalkylamine acetylation in the *PAA1* overexpressing strain could be explained because arylalkylamines are not the main in vivo substrate of Paa1p, although it conserves a lower specific activity than that of the mammalian enzyme [15]. Liu et al. [18] also provided strong evidence that spermine was the main in vivo substrate of Paa1p. However, the fact that we also detected acetylated activity in the null paa1 mutant suggests that there may be other N-acetyltransferases in *S. cerevisiae*. Another explanation of the obtained results is that, in spite of the overexpression, this higher transcriptional activity in *PAA1* is not enough to increase the concentration of acetylated arylalkylamines. As it is well known that *E. coli* produces higher expression levels of recombinant proteins than *S. cerevisiae*, and several previously uncharacterized members of the yeast *N*-acetyl transferases were expressed in *E. coli*, and the recombinant proteins were purified and assayed for their acetylation activity [15,19], we decided to overexpress *PAA1* in *E. coli* and determine the acetylation activity in different arylalkylamines, in a similar in vivo bioconversion assay to the one performed in *S. cerevisiae*.

### 3.2. Overexpression of PAA1 in E. coli

In order to assess *PAA1* activity in *E. coli*, an in vivo bioconversion of 5-methoxytryptamine into melatonin was performed in an inducible overexpression system and, conversely to the overexpression in *S. cerevisiae*, a significant increase in melatonin production was detected in comparison to the wild-type strain (Figure 2A). As previously, the Bt*AANAT* was also overexpressed as a positive control and this system yielded much higher titers of melatonin than the strain overexpressing *PAA1* (more than 20-folds), evidencing a much higher specific activity of the mammalian enzyme. Once the overexpression of *PAA1* showed positive bioconversion results, we tested Paa1p acetylase activity for other possible substrates related to the melatonin pathway, namely tryptamine and serotonin, to test *PAA1* specificity and preference of substrate by measuring the corresponding acetylated products *N-*acetyltryptamine and *N-*acetylserotonin. Interestingly, we observed in vivo acetylase activity of *PAA1* using tryptamine, 5-methoxytryptamine, and serotonin as a substrate, being tryptamine the preferred substrate that yielded approximately 1 µg·mL^−1^ of *N-*acetyltryptamine, while serotonin produced the lowest amount of acetylated product with 50 ng·mL^−1^, which was still significant when compared to the wild-type strain (Figure 2B).

With all the evidence obtained with the overexpression of *PAA1* in both host organisms, we reasoned *PAA1* might be involved in the melatonin biosynthetic pathway in yeast, but it is not essential for an in vivo significant production of this compound. This makes clear there are still other candidates to consider as responsible for the acetylase step in the melatonin pathway, and probably, as well as for *PAA1*, they may have other main functions. The fact melatonin is produced in such small amounts in yeast, and the enzymes responsible for the different necessary reactions are not exclusive of this pathway, highlights the difficulty of the search for candidates using yeast cells for the bioconversion assays. Thus, we concluded bioconversion assays for testing possible candidates should be carried out using the bacterial expression system because they produce an outsized effect of the tested protein, revealing any possible acetylating activity, even when it happens in a residual way in yeast.

### 3.3. Search for New N-Acetyltransferases in S. cerevisiae

Melatonin biosynthesis in yeast is not a conspicuous trait in terms of the amount of generated product, but its importance relies on melatonin’s free radical scavenging activity and its modulation of gene expression, even when it is present at low concentrations [20,21]. All gathered evidence around the *PAA1* gene, together with the usual low concentrations of melatonin detected as a result of spontaneous biosynthesis, led us to infer that enzymes involved in this biosynthetic route, and especially *PAA1*, may not be exclusive for this route but they rather have other main functions instead, even though they can eventually contribute to melatonin biosynthesis in a leaky and inefficient manner. The search for these possible gene candidates involved in melatonin synthesis becomes a complicated and subtle labor in which expression levels of specific genes can bring out their relevance in this process. As we understand, a global transcriptome analysis during melatonin production can provide an interesting starting point in the search for candidate genes, especially those involved in the acetyltransferase activity, if a melatonin synthesis situation triggers a differential expression of the genes involved in it. Despite multiple pieces of evidence of melatonin function in *S. cerevisiae*, the external conditions that are capable of inducing its production are difficult to establish as no media and growth conditions have been unequivocally associated with a spontaneous increase in detectable melatonin levels. For this reason, following our previous approach for a reproducible melatonin synthesis [14], we directly induced this synthesis by supplementing the growth media with an immediate precursor such as 5-methoxytryptamine to observe melatonin production (Figure 3A).

We explored transcriptional response after 15 and 45 min of the precursor supplementation and, under our strict criteria for significance, only 13 and 8 genes were significantly upregulated at times of 15 and 45 min, respectively. We observed an intense activation of genes related to iron and copper homeostasis and specific genes related to transmembrane transporter activity, especially after 15 min of the precursor addition, when we compared supplemented culture with its non-supplemented control (Figure 3B). A reasonable explanation for this result is that melatonin and some precursors such as 5-methoxytryptamine can act as iron and copper chelators, as previously pointed out [20,22,23]. Therefore, a change in metal ion availability in the growth medium may occur when using 5-methoxytryptamine as a supplement to produce melatonin. Differences in media composition regarding heavy metals between our supplemented and control cultures need to be taken into account when analyzing expression patterns as a certain grade of the transcriptional response to metal deficiency is inevitably expected. For a time of 45 min, the overexpressed genes belong to the “protein folding, and protein targeting to ER” network cluster (STRING), indicating the response to chelation occurred in a short period of time, and the main overexpression at 45 min is related to replicative and translational stress response (Figure 3C).

We believe the response to the chelating effect of the studied compounds has apparently masked any less conspicuous gene expression and hindered their significance. Therefore, in order to be able to detect other activated genes in response to 5-methoxytryptamine supplementation or melatonin production, we looked further down in the list of overexpressed genes which are structurally related to *AANAT* and cautiously considered them as possible candidates. We used RNAseq information as a guide and crossed it with the InterPro database [24] to search for protein candidates among *S. cerevisiae* with a functional homology with the GNAT domain (IPR000182) of the reference gene *AANAT*. We found eight coincidences between the family domain search results and overexpressed genes from the RNAseq results, one of them expectedly being *PAA1*, so we considered seven new gene candidates with a GNAT domain for testing for melatonin production (Table 1).

### 3.4. Detection of AANAT Activity in the New Gene Candidates by Overexpression in E. coli

The seven selected candidates were overexpressed in *E. coli* as described above. Six out of these seven genes did not show any significant difference in the melatonin yielded in comparison with the wild type. Thus, the involvement of these genes in the putative production of melatonin or in the acetylation of other compounds of the route can be ruled out. Nonetheless, we detected that the overexpression of the gene *HPA2* resulted in significantly higher titers of melatonin, in an amount similar to the previously reported for *PAA1* and, again, in lower concentration than the production of the overexpression of Bt*AANAT* (Figure 4). The new positive candidate *HPA2* was early overexpressed at a time of 15 min, while *PAA1* overexpression was detected only in the next time point analyzed at 45 min (Figure 5). The sequential transcription of different genes involved in melatonin synthesis can explain the lack of a strong transcriptional response of one single responsible gene, reinforcing the theory of a common function from multiple genes when it comes to melatonin biosynthesis. *HPA2* is a tetrameric histone acetyltransferase, a member of the Gcn5 acetyltransferase family, which acetylates histones H3 and H4 in vitro and also acetylates polyamines. However, this is the first report that clearly proves the involvement of this enzyme in the acetylation of arylalkylamines, such as 5-methoxytryptamine, and, therefore, in the synthesis of melatonin in yeasts. As was the case for *PAA1*, the substrate specificity and the specific activity was significantly lower than the mammalian enzyme, but, likely, as a moonlighting protein, Hpa2 was able to significantly convert 5-methoxytryptamine into melatonin.

## 4. Conclusions

So far, only the *PAA1* gene has been correlated with the unknown pathway of melatonin biosynthesis in yeasts. We aimed to characterize the enzymatic activity of this gene by whole-cell biotransformation. To this end, we overexpressed *PAA1* in its own host, *S. cerevisiae*, and in *E. coli*. However, we did not detect significant acetylation activity in *S. cerevisiae* whereas this higher activity was evident in *E. coli*. Therefore, a clear conclusion for the future search for new enzymes of the route is that the overexpression in *E. coli* reveals higher differences in the enzymatic activity. Our results also evidenced that *PAA1* was not the only enzyme with AANAT activity in *S. cerevisiae*. The combination of criteria from transcriptomics and structure prediction to find similar domains let us narrow down the list of gene candidates to test in an in vivo assay that resulted in the proposal of a new gene candidate. We can conclude that *HPA2*, a histone acetyltransferase also related to polyamine acetylation, was able to significantly convert 5-methoxytryptamine to melatonin. Therefore, together with Paa1, Hpa2 should be also considered an arylalkylamine *N-*acetyltransferase in *S. cerevisiae*. However, both enzymes should be considered moonlighting proteins, and taking into account the yield of acetylated arylalkylamine in comparison with the mammalian enzyme, this AANAT activity does not seem to be the main acetylation activity. Therefore, the presence of an AANAT enzyme with a higher arylalkylamine substrate specificity in *S. cerevisiae* cannot be ruled out. We are currently applying new and powerful bioinformatic tools for the search for the most specific AANAT enzymes.

## Figures and Tables

**Figure 1 microorganisms-11-01115-f001:**
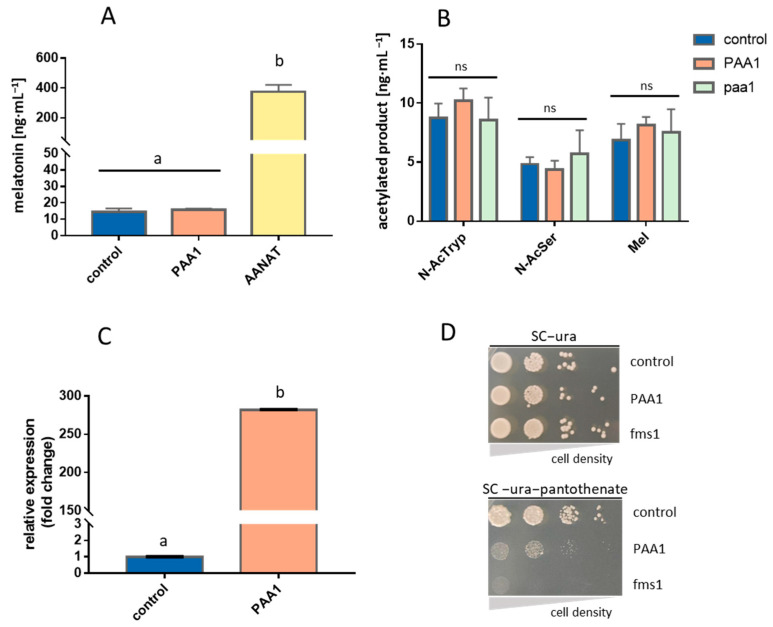
In vivo testing of *PAA1* function in yeast strain BY4743. (**A**) Bioconversion assay for melatonin production on overexpressing strains. (**B**) Yeast in vivo production of *N*-acetyltryptamine (*N-*AcTryp), *N*-acetylserotonin (*N-*AcSer), and melatonin (Mel) was achieved by supplementing precursors tryptamine, serotonin, and 5-methoxytryptamine, respectively, either in overexpressing (PAA1) or null mutant (paa1) strain. (**C**) An effective overexpression of *PAA1* resulted in a great increase in mRNA levels when compared to control strain. (**D**) *PAA1* is functional as it increases the consumption of polyamines as substrates and provokes a growth defect when no pantothenate is available. This growth defect reflects depletion of polyamine precursors of pantothenate due to the increased action of Paa1p. p426GPD backbone with no cloned gene was used as a control in all cases. Different letters “a, b” indicate groups that are significantly different (*p* < 0.05). “ns” reflects no significant difference when comparing the mean values below.

**Figure 2 microorganisms-11-01115-f002:**
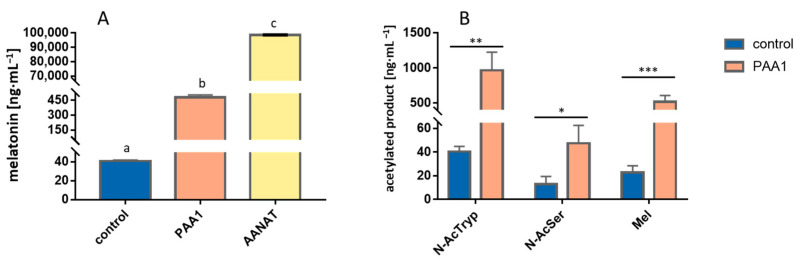
In vivo testing of *PAA1* function in *E. coli* by bioconversion assays showing a significant acetylation activity for melatonin production when supplemented with 5-methoxytryptamine (**A**), but also for *N-*AcTryp and *N-*AcSer when supplemented with tryptamine and serotonin, respectively, demonstrating a broad substrate scope of this gene (**B**). pGEX-5X-1 backbone without any cloned gene was transformed into *E. coli* and used as a control in these assays. Different letters “a–c“ indicate variables that are significantly different from each other (*p* < 0.05). Asterisks show significant differences in relation to their control (*p* < 0.05 (*), *p* < 0.005 (**), and *p* < 0.001 (***)).

**Figure 3 microorganisms-11-01115-f003:**
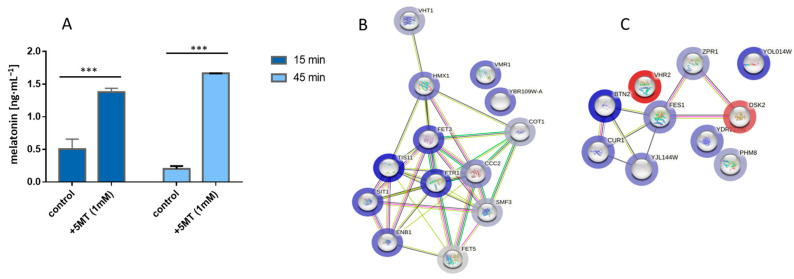
Melatonin production modulated yeast transcriptional response. (**A**) Melatonin production achieved by supplementing the media with precursor 5-methoxytryptamine (5MT). Samples were taken at 15 and 45 min after the 5MT addition are shown. Transcriptional response is depicted as a network of predicted interaction between the different genes (nodes) showing whether they are overexpressed (blue rim) or repressed (red rim) relative to control at time 15 min (**B**) or 45 min (**C**). Asterisks show significant differences in relation to their control (*p* < 0.001 (***)).

**Figure 4 microorganisms-11-01115-f004:**
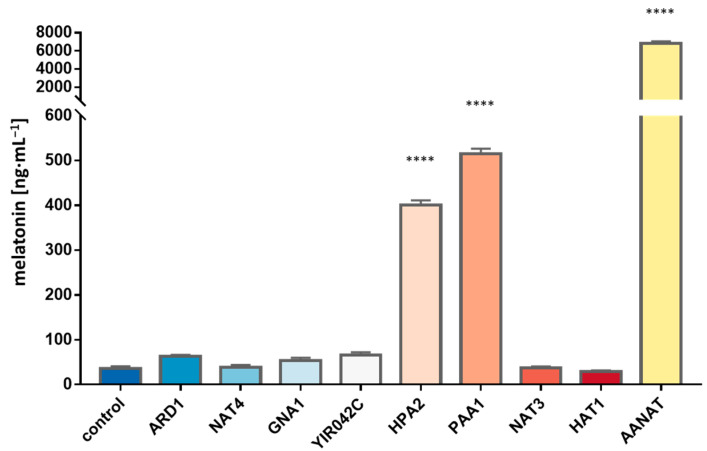
Melatonin production from 5-methoxytryptamine of the different gene candidates extracted from RNAseq results and homology domain search results. Genes were overexpressed in *E. coli* and pGEX-5X-1 backbone with no cloned gene was used as a control. Asterisks show significant differences in relation to the control (*p* < 0.0001 (****)).

**Figure 5 microorganisms-11-01115-f005:**
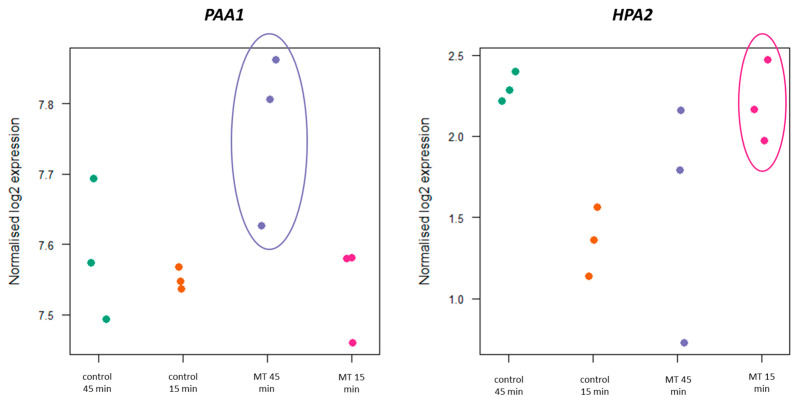
Sequential overexpression of *HPA2* and *PAA1* under melatonin synthesis conditions. Different samples are depicted as colored dots where each color belongs to a combination of treatment (control or supplemented) and sampling time (15 or 45 min). RNAseq results showed expression levels of the 5-methoxytryptamine supplemented samples (MT) for *HPA2* were higher than control at time 15 min (*p*-value: 0.0185), while *PAA1* showed higher transcript levels at time 45 min (*p*-value: 0.0319).

**Table 1 microorganisms-11-01115-t001:** Plasmids used in this study.

Name	Relevant Traits	Integration Site/Replicon	Marker	Source
pGEX-5X-1	Taq promoter-GST-MCS	pBR322 (ColE1)	Amp	GE Healthcare
pGEX-5X-1 PAA1	Taq promoter-GST-*PAA1*	ColE1	Amp	This study
pGEX-5X-1 AANAT	Taq promoter-GST-Bt*AANAT*	ColE1	Amp	This study
pGEX-5X-1 ARD1	Taq promoter-GST-*ARD1*	ColE1	Amp	This study
pGEX-5X-1 NAT4	Taq promoter-GST-*NAT4*	ColE1	Amp	This study
pGEX-5X-1 GNA1	Taq promoter-GST-*GNA1*	ColE1	Amp	This study
pGEX-5X-1 YIR042C	Taq promoter-GST-*YIR042C*	ColE1	Amp	This study
pGEX-5X-1 HPA2	Taq promoter-GST-*HPA2*	ColE1	Amp	This study
pGEX-5X-1 NAT3	Taq promoter-GST-*NAT3*	ColE1	Amp	This study
pGEX-5X-1 HAT1	Taq promoter-GST-*HAT1*	ColE1	Amp	This study
p426GPD	*TDH3*p-MCS-*CYC1*t	2μ	URA3	[17]
pCfB2628	TEFp-Hs*DDC*, PGK1p-Bt*AANAT*	XI-5	HIS5	[16]
p426GPD PAA1	*TDH3*p-*PAA1*-*CYC1*t	2μ	URA3	This study
p426GPD AANAT	*TDH3*p-Bt*AANAT*-*CYC1*t	2μ	URA3	This study

**Table 2 microorganisms-11-01115-t002:** Primers used in this study.

Title 1	Title 2
PAA1 F BamHI	AGGTCGTGGGATCCCCATGGCCTCCTCAAGTAGCA
PAA1 R XhoI	TGCGGCCGCTCGAGCTAGTTGTCGTATTCTTCCTTAAT
AANAT F BamHI	AGGTCGTGGGATCCCCATGAGCACCCCGAGCAT
AANAT R XhoI	TGCGGCCGCTCGAGTTAACGATCGCTATTACGACGCA
GPD Pro F	CGGTAGGTATTGATTGTAATTCTG
CYC1-R	GCGTGAATGTAAGCGTGAC
pGEX seq F	GGGCTGGCAAGCCACGTTTGGTG
pGEX seq R	CCGGGAGCTGCATGTGTCAGAGG
PAA1 (qPCR) F	GGTTTCCCACCAAACGAAAG
PAA1 (qPCR) R	ACTTCTTTGCCCTCGATCTC

## Data Availability

The data presented in this study are available in Appendix A.

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
