# Peer review of "The Role of the PAA1 Gene on Melatonin Biosynthesis in Saccharomyces cerevisiae: A Search of New Arylalkylamine N-Acetyltransferases"

_microorganisms, 2023, doi:10.3390/microorganisms11051115_

Round 1

Reviewer 1 Report

It is interesting a topic to elucidate the melatonin biosynthesis pathway in Saccharomyces cerevisiae. However, the manuscript is not well prepared as there are many to work on.
1. All the Latin names of the microbes should be in italic.
2. The manuscript should be reconstructed. The authors aimed on investigated the gene on melatonin biosynthesis in Saccharomyces cerevisiae, and they studied the genes in E. coli. However, they didn't mention the heterologous results in the abstract. 
3. To verify the function of PAA1 on melatonin biosynthesis, the gene should be deleted to give direct evidence.
4. The structure seems somewhat confusing. The authors put their results in a crosstalk order. Why didn't the authors first give a thorough investigation over the native genes of interest in S. cerevisiae and then verify them together in E. coli.
5. The conclusion should be take-home message. In the present form, the conclusion part is much too long, and has some content should be considered as discussion.

Author Response

Responses to Reviewer 1

It is interesting a topic to elucidate the melatonin biosynthesis pathway in Saccharomyces cerevisiae. However, the manuscript is not well prepared as there are many to work on.
1. All the Latin names of the microbes should be in italic.

We have checked all the Latin names of the microbes and all of them are now italics. We thank the reviewer’s comment because we missed a few of them.

  1. The manuscript should be reconstructed. The authors aimed on investigated the gene on melatonin biosynthesis in Saccharomyces cerevisiae, and they studied the genes in E. coli. However, they didn't mention the heterologous results in the abstract. 

We are a bit confused with this comment because we consider that we followed this logical order in the manuscript. As it is mentioned, we firstly characterized the in vivo function of PAA1 by overexpressing this gene in S. cerevisiae. However, as we did not observe significant differences in melatonin production in comparison to the wild type strain or the null paa1 mutant, we decided to overexpress PAA1 in E. coli as it is well-known that E. coli produces higher expression levels of recombinant proteins than S. cerevisiae. Conversely to S. cerevisiae, the overexpression of PAA1 in E. coli showed a significant increase in melatonin production in comparison to the wild type strain. In the light of this result, we concluded that bioconversion assays for testing possible candidates should be carried out using the bacterial expression system because they produce an outsized effect of the tested protein, revealing any possible acetylating activity, even when it happens in a residual way in yeast.

In agreement to this conclusion, in section 3.4, and after the search for new N-acetyltransferases in S. cerevisiae, we decided to only show the results of the overexpression of these candidate genes in E. coli, in which the gene HPA2 also evidenced N-acetyl transferase activity together with PAA1.  Anyway, we also tested the overexpression of these candidate genes in S. cerevisiae, however, as expected, we did not detect significant differences in melatonin production in any of the overexpressing strain and the wild type. These results are plotted in the next graph:

(see in the attached document)

Figure: Melatonin production from 5-methoxytryptamine of the different gene candidates overexpressed in S. cerevisiae

However, we decided not to show these data in the manuscript because are completely irrelevant and only confirmed our previous conclusion that E. coli is a better expression system than its own host S. cerevisiae to test candidate genes of this biochemical pathway.

Finally, we agree with the reviewer that this conclusion is an important result of the manuscript and should be mentioned in the Abstract. We have now included the next sentence in the Abstract:

“The AANAT activity of the candidate genes was validated by their overexpression in E. coli because, curiously, this system evidenced higher differences than the overexpression in their own host S. cerevisiae

  1. To verify the function of PAA1 on melatonin biosynthesis, the gene should be deleted to give direct evidence.

We again detect some misunderstanding in this reviewer comment because null mutant for gene PAA1 has been used in bioconversion assays in S. cerevisiae, and yielded concentrations of melatonin similar to control and overexpressing strain (Figure 1B and lines 228-235).

As we explained in lines 272-274, the fact that we also detected acetylated activity in the null paa1 mutant suggests that there may be other N-acetyltransferases in S. cerevisiae. This result prompted us to search new genes with N-acetyltransferase activity (section 3.3)

  1. The structure seems somewhat confusing. The authors put their results in a crosstalk order. Why didn't the authors first give a thorough investigation over the native genes of interest in S. cerevisiae and then verify them together in E. coli.

We have already responded to this comment in query 2

  1. The conclusion should be take-home message. In the present form, the conclusion part is much too long, and has some content should be considered as discussion.

Following the reviewer’s recommendation, we have shortened the conclusion section by moving some of the paragraphs to the “Result and Discussion” section

Reviewer 2 Report

The article " The role of PAA1 gene on melatonin biosynthesis in Saccharomyces cerevisiae: a search of new arylalkylamine N-acetyltransferases" is devoted to the study of melatonin biosynthesis in yeast cells and the characterization of the Paa1 enzyme, which was previously considered a homolog of the vertebrate’s aralkylamine N-acetyltransferase (AANAT).

Given the diverse functions of melatonin, as well as differences in the melatonin biosynthesis pathway in mammals, plants, and microorganisms, the results obtained by the authors are of undoubted interest.

Obtaining yeast and bacterial strains with overproduction of the yeast enzyme Paa1 allowed the authors to evaluate the activity and substrate specificity of this enzyme in vivo and prove that arylalkylamines are not the main in vivo substrate of Paa1p and suggest that there may be other N-acetyltransferases in S. cerevisiae. The use of transcriptome analysis allowed the authors to identify another protein with AANAT activity - Hpa2. As is known. Hpa2 is a histone acetyltransferase, the results obtained expand the understanding of the function of this enzyme in yeast.

One question:

1. Line 186: Is everything correct in this sentence (the word treated was used twice) “The RNA used in the qPCR analysis was treated with DNase treated for 15 minutes at 25°C…”?

Author Response

Responses to Reviewer 2

The article " The role of PAA1 gene on melatonin biosynthesis in Saccharomyces cerevisiae: a search of new arylalkylamine N-acetyltransferases" is devoted to the study of melatonin biosynthesis in yeast cells and the characterization of the Paa1 enzyme, which was previously considered a homolog of the vertebrate’s aralkylamine N-acetyltransferase (AANAT).

Given the diverse functions of melatonin, as well as differences in the melatonin biosynthesis pathway in mammals, plants, and microorganisms, the results obtained by the authors are of undoubted interest.

Obtaining yeast and bacterial strains with overproduction of the yeast enzyme Paa1 allowed the authors to evaluate the activity and substrate specificity of this enzyme in vivo and prove that arylalkylamines are not the main in vivo substrate of Paa1p and suggest that there may be other N-acetyltransferases in S. cerevisiae. The use of transcriptome analysis allowed the authors to identify another protein with AANAT activity - Hpa2. As is known. Hpa2 is a histone acetyltransferase, the results obtained expand the understanding of the function of this enzyme in yeast.

One question:

  1. Line 186: Is everything correct in this sentence (the word treated was used twice) “The RNA used in the qPCR analysis was treated with DNase treated for 15 minutes at 25°C…”?

It was not correct because it was a mistake in the wording that have been amended in the revised version. Thanks for the comment.

Round 2

Reviewer 1 Report

All my concerns have been well addressed.